# Strategies to Improve the Properties of Amaranth Protein Isolate-Based Thin Films for Food Packaging Applications: Nano-Layering through Spin-Coating and Incorporation of Cellulose Nanocrystals

**DOI:** 10.3390/nano10122564

**Published:** 2020-12-21

**Authors:** Amparo López-Rubio, Adriana Blanco-Padilla, Kristiina Oksman, Sandra Mendoza

**Affiliations:** 1Preservation and Food Safety Technologies, IATA-CSIC, Avda. Agustin Escardino 7, 46980 Paterna, Spain; 2Departmento de Investigación y Posgrado en Alimentos, Facultad de Química, Universidad Autónoma de Querétaro, Querétaro 76010, Mexico; maloru1@hotmail.com (A.B.-P.); smendoza@uaq.mx (S.M.); 3Division of Materials Science, Luleå University of Technology, SE-97187 Luleå, Sweden; kristiina.oksman@ltu.se

**Keywords:** amaranth protein isolate, cellulose nanocrystals, spin coating, thin films, mechanical properties, food packaging

## Abstract

In this work, two different strategies for the development of amaranth protein isolate (API)-based films were evaluated. In the first strategy, ultrathin films were produced through spin-coating nanolayering, and the effects of protein concentration in the spin coating solution, rotational speed, and number of layers deposited on the properties of the films were evaluated. In the second strategy, cellulose nanocrystals (CNCs) were incorporated through a casting methodology. The morphology, optical properties, and moisture affinity of the films (water contact angle, solubility, water content) were characterized. Both strategies resulted in homogeneous films with good optical properties, decreased hydrophilic character (as deduced from the contact angle measurements and solubility), and improved mechanical properties when compared with the neat API-films. However, both the processing method and film thickness influenced the final properties of the films, being the ones processed through spin coating more transparent, less hydrophilic, and less water-soluble. Incorporation of CNCs above 10% increased hydrophobicity, decreasing the water solubility of the API films and significantly enhancing material toughness.

## 1. Introduction

The use of biopolymers, such as polysaccharides, lipids, or proteins, has generated growing interest in recent years for their potential application as edible films and coatings. Specifically, these sustainable materials can be interesting to replace synthetic plastics in food packaging applications, thus helping to counteract the problem of plastic pollution [1,2]. Various types of proteins such as wheat gluten, soy, gelatin, corn zein, peanut, sunflower, milk, and whey proteins have been evaluated as potential film-forming agents [3,4,5]. Although the developed materials have good barrier properties to oxygen, flavors, and lipids, they show high water vapor permeability and poor mechanical properties [3,5].

Amaranth (*Amaranthus hypochondriacus*) is a pseudo-cereal with high protein content (17%) and an amino acid composition close to the optimum balance required by human nutrition [6]. Furthermore, due to its low production cost, amaranth protein has begun to be used as biopolymer in the development of micro and nanostructures for the encapsulation of active compounds [7,8,9] or as edible films [10,11]. In the case of the amaranth-based protein films, poor mechanical and barrier properties have been observed [10], thus highlighting the need to find different strategies in order to improve these characteristics.

The most frequently used strategies to enhance the properties of the protein-based films are the use of biopolymer blends, coating with materials with high barrier properties, or fabrication of multilayer materials, among others [12]. Another strategy to improve the mechanical and barrier properties of protein films is the use of nanoreinforcements [13] to generate the so-called nanocomposites. Natural polysaccharides, such as nano-sized cellulose, chitin, and starch, can be employed as nanoreinforcing elements because of their high mechanical properties [14]. These materials are semicrystalline and consist of crystalline and amorphous regions. The amorphous regions are susceptible to hydrolysis leaving only the crystalline fraction, which is normally called cellulose nanocrystals, chitin nanowhiskers, and platelet-like starch nanocrystals, respectively [15]. These crystalline carbohydrate nanoparticles, apart from being impermeable to the passage of low molecular weight molecules like gases, constitute rigid elements, thus generally providing improved mechanical properties to the bulk biopolymer matrix. Cellulose nanocrystals (CNCs) have been extensively used in polymer nanocomposites to improve mechanical and barrier properties or swelling behavior [16,17]. The addition of CNCs from different sources to protein films such as gelatin [18], wheat gluten [19], and soy [20] has resulted in the improved barrier and mechanical properties of the resulting materials. Condés et al. [21] evaluated the effect of the addition of starch nanocrystals in amaranth protein films, and the results showed that the inclusion of starch nanocrystals improved the water uptake and mechanical behavior of amaranth protein films.

Although the casting method is a widely employed technique for the development of biopolymer films [22], spin-coating is a well-established technology that allows the production of ultrathin and uniform films onto flat substrates [23]. The process involves the deposition of a small amount of a solution on the center of the substrate, which is then rotated at high speed to spread the coating material by centrifugal force, thus leaving a thin film of material on the surface of the substrate. The application of this technique has been mostly focused on the electronic industry [24], but it can be envisaged as an interesting technique to develop multilayer packaging structures or active coatings for food packaging structures. For instance, this technique has been used to develop multilayer complexes based on poly (lactic acid) and wheat gluten with improved barrier properties [25], or to prepare nanocellulose coatings that displayed low oxygen permeability at low relative humidity conditions [26]. However, the use of this processing technique for pure protein thin-film development has not been explored. Moreover, differences in film properties arising from the production methods are also worthwhile to be explored.

The aim of this work was to prepare and characterize films based on amaranth protein isolate (API) using two different strategies, i.e., nanolayering of the API through spin coating and CNCs addition to develop API nanocomposite films through casting. Moreover, given the scarce studies in this area, the aim of the present study was to evaluate the spin coating conditions to produce API films of interest in food packaging applications. The effects of protein concentration, rotation speed and number of layers on the morphology, optical properties, moisture affinity and mechanical properties of the films were evaluated.

## 2. Materials and Methods

### 2.1. Materials

The commercial amaranth protein concentrate (*Amaranthus hypochondriacus* L. Revancha variety) was supplied by Nutrisol (Hidalgo, Mexico). The amaranth protein isolate (API) was prepared based on the methodology previously reported [27]. Briefly, a water suspension of defatted commercial amaranth protein concentrate was adjusted to pH 9 with a 2 M NaOH. The mixture was centrifuged after stirring 30 min at room temperature. Then, the supernatant was adjusted to pH 5 with 2 M HCl, centrifuged at 4 °C, and the pellet was resuspended in water, adjusted at pH with 0.1 M NaOH, and freeze-dried. The protein content of the API was 85.5 ± 0.2% and consisted of a mixture of different proteins with molecular weights ranging from ~10 to ~83 kDa [27]. Sodium hydroxide (NaOH) was purchased from VWR International AB (Stockholm, Sweden). Formic acid (HCOOH) of 95% purity, glycerol and the surfactant Tween 80 were supplied by Sigma-Aldrich Co. (St. Louis, MO, USA). Freeze-dried CNCs (2012-FPL-CNC-043) were kindly supplied by USDA Forest Products Laboratory, Forest Service (Madison, WI, USA). The CNCs were produced from wood pulp using sulfuric acid and the dimensions were between 100–300 nm in length and around 5 nm diameter [28]. Milli-Q water was used as a solvent.

### 2.2. Preparation of the API Solutions

The API solutions were prepared by dissolving three different concentrations of the protein isolate (5%, 10%, and 15% w/v) in 95% formic acid. The solutions were heated to 70 °C and mixed for 10 min at 250 rpm, using magnetic stirring (VMW-C10, VWR, Darmstadt, Germany); the stirring continued at room temperature until complete dissolution. After the dissolution, 25 wt% glycerol as plasticizer (relative to the API weight) was added into the solution and stirred another 10 min at room temperature. For the cast films, variable amounts of CNCs (0, 5, 10, and 20 wt% relative to protein isolate mass) were prepared in distilled water and incorporated into the 5% API solutions.

### 2.3. Nano-Layering through Spin Coating Process

The spin coating was performed using a Brewer Science Inc. 200X (Rolla, MO, USA). Five hundred microliters of each solution were placed on the acetate sheet used as a substrate. The substrate was rinsed three times with the constant addition of ethanol and deionized water prior to the spin coating process. All experiments were performed at room temperature.

A Taguchi method was used to evaluate the spin coating conditions limiting the number of trials. Specifically, an L_9_ orthogonal array was chosen for studying the effects of three factors at three levels on the important properties of the films developed. In Table 1, the range of different process parameters and factor levels used for this study are compiled, while Table 2 shows the experiments conducted following the Taguchi method. The number of layers were set to have enough variability of film thicknesses.

### 2.4. Solvent Casting

All dispersions of 5% API containing the CNCs (0, 5, 10, and 20 wt% relative to protein isolate mass) were magnetically stirred for 1 h at room temperature, their pH was adjusted to 10.5 with 2 mol/L NaOH, and they were stirred again for 24 h and sonicated for 10 min. 10 g of each film-forming suspension were poured onto polystyrene Petri dishes (60 mm diameter) and dried at 40 °C for 4 h in an oven and were subsequently left 24 h at room temperature. The dry films were conditioned at 58% relative humidity in a desiccator with a saturated solution of sodium chloride for at least 48 h before the characterization.

### 2.5. Characterization

Film thickness was measured with a hand-held micrometer (Mitutoyo ID-C112XB, Okinawa, Japan). The thickness at ten locations in three different samples was obtained and expressed as the mean thickness of the corresponding film.

The optical transparency of the films was studied using a Perkin Elmer UV/Vis Spectrometer Lambda 2S (Überlingen, Germany) with a resolution of 0.2 nm. For that purpose, the light transmittance of the films in a light wavelength range from 350 to 800 nm was measured. The scan speed used in the analysis was 240 nm/min, and four replicates of each material were measured. Transparency of the films was calculated using the following equation [29]:Transparency = A_600_/*X*(1)
where A_600_ is the absorbance at 600 nm and *X* is the film thickness (mm).

The morphology of the cross-sections from the nanocomposite films was studied using scanning electron microscopy (SEM) JEOL (JSM-IT300, Tokyo, Japan) at an accelerating voltage of 15 kV and a working distance of 22 mm. The samples were sputter-coated with tungsten under vacuum before prior to examination.

Contact angle measurements were determined using a dynamic absorption method [30]. Briefly, one drop of water (4 µL) was placed by contact angle tester fibro dat 1120 FIBROSystem AB (New Castle, UK) onto the coated acetate surface, and the contact angle measurements at 1 s were used for the comparisons. Six different measurements were made in each type of film developed.

Water solubility was measured as reported by Leceta et al. [31]. Briefly, the specimens were placed in a flask with 25 mL of distilled water. The flasks were stored at room temperature for 24 h. After this time, the specimens were dried in an oven at 105 °C for 24 h. The water solubility was calculated in relation to the dry mass and was expressed as the percentage of film dry-matter solubilized.

Moisture content (MC) was also determined as previously reported [28]. Film samples were weighed (w1), dried in an oven at 105 °C for 24 h, and weighted (w2) again. WC was determined as the percentage of initial film-weight lost during drying and reported on a wet basis. Three measurements of MC were obtained for each type of film, and an average was taken as the result. Total soluble matter (TSM) was measured as previously reported [31]. Briefly, the specimens were placed in a flask with 25 mL of distilled water. The flasks were stored at room temperature for 24 h. After this time, specimens were dried in an oven at 105 °C for 24 h. TSM was calculated in relation to the dry mass and was expressed as the percentage of dry film matter solubilized.

Mechanical properties were evaluated using a conventional tensile testing machine Shimadzu AG-X (Kyoto, Japan) at room conditions (21 °C and RH 35%) following ASTM D638 standard. Samples were conditioned for two days before the testing. The testing was done using 1 kN load cell, gauge length 20 mm and the crosshead speed was of 2 mm/min. The tensile strength and elongation at the break were directly determined from the stress-strain data, and Young’s modulus was calculated as the steepest slope of the initial linear part of the curve. The results are average on four measurements. One-way analysis of the variance (ANOVA) was performed using XLSTAT-Pro (Win) 7.5.3 (Addinsoft, NY, USA) software package. Differences between samples were evaluated using the Tukey test (α = 0.05).

## 3. Results and Discussion

### 3.1. Optical Properties and Microstructure

Figure 1 shows the appearance of the amaranth-based films obtained following both methodologies (casting and spin-coating). The difference in tonality observed was mainly explained by the thickness of the films. While the thickness of cast films ranged between 76 to 94 µm and they had a brownish color provided by the amaranth protein, spin coating generated much thinner films with thicknesses below 30 µm (cf. Table 3), giving rise to much more transparent films in general. The exceptions were the spin-coated films obtained from the 15% protein solution with 50 and 100 layers and, especially, the latest one, which was obtained at a low spinning rate, and showed a rather heterogeneous appearance and varying thickness, as less material was deposited in the center of the film due to the centrifugal force during the film-forming process. The three factors considered in the Taguchi experimental design (protein content, rotational speed, and the number of layers) had a significant effect on film thickness. These results are consistent with those reported by Hall et al. [32], who suggested that the rotational speed was inversely proportional to film thickness. Regarding the nanocomposite films of amaranth obtained by casting and containing different concentrations of cellulose nanocrystals (CNC), they were homogenous, translucent, and slightly brownish with a visual appearance similar to the control film (without CNCs) at a macroscopic scale.

The proper incorporation of CNC within the cast films was assessed by SEM, and the images are shown in Appendix A. The cross-sections of the films displayed a homogeneous microstructure, and no signs of CNC agglomeration or phase separation could be observed, thus confirming a homogenous distribution of the nanofillers and good compatibility with the protein matrix.

The optical properties of biopolymer-based films depend on the nature of the biopolymer and plasticizer used in their production, as well as the production method [13]. Figure 2 shows light transmission in the UV-visible spectra for the different films developed. In accordance with the visual appearance of the films, greater light transmission values were observed for the spin-coated films in the whole spectra. It is interesting to note the ability of cast films to block UV light, which could be useful for certain applications, like the packaging of light-sensitive products. As observed from Figure 2A, incorporation of CNC did not affect the transparency of the films, and the differences observed between the various formulations are probably due to the uneven distribution of the protein, which accumulates in certain areas of the film (more intense yellow color).

In the case of the spin-coated films (Figure 2B), no significant differences in the transmittance of the films was observed, being greater than 80% between 400–600 nm for almost all films. Again, the exception was the film produced from the API solution with the greatest concentration (15%) and greater number of layers (100), which, as commented before, had a rather heterogeneous appearance and uneven material distribution. These values were similar to those reported in a previous study [27] for pullulan and pullulan-whey protein isolate films with ~69–88% and ~81–88% transmittance values, respectively, but higher than those obtained by Shiku et al. [33] for fish myofibrillar protein films with transmittance values of ~51–81%. The results indicated that API films were clear enough to be used as see-through coating and/or packaging material.

Table 3 compiles the transparency values of all the films, showing that most of them had acceptable values comparable to polymer films commonly used in food packaging, such as low-density polyethylene (3.05), oriented polypropylene (1.67), and polyester (1.51) [33]. Interestingly, when comparing the films prepared by casting and spin-coating having the same protein concentration (5%), the latter ones displayed greater transparency values. Although they cannot be directly compared as thicker films were obtained through casting, a fact which could influence film color, the results seem to point out that this processing technology may lead to more transparent films. The transparency results were comparable or even better than those previously reported for Dosidicus gigas muscle protein films [34] and for whey protein films [35] with reported transparency values between 0.73–3.48 and 1.11–3.43, respectively.

### 3.2. Moisture Sensitivity of the API Films

Another important property when developing films for food packaging applications is their water sensitivity. One way of characterizing the hydrophilicity of biopolymer films is through their water contact angle. Table 3 also includes the contact angle values for the different amaranth-based materials developed. In general, in the spin-coated films, the water contact angle on the surface of API films decreased as the protein concentration increased, indicating that the films with greater protein content were the most hydrophilic ones. The same trend was observed in films with different amounts of gelatin, where it was seen that the water contact angle decreased as the amount of gelatin present in the film was increased [36]. Higher contact angles of ~68 have been reported for soy protein films [37] while a contact angle of ~18 was reported for sunflower protein isolate films produced by thermo-molding [38]. Again, reviewing these results shows that both the processing and film thickness have an important influence on the films’ final properties, as compared to the control cast film with those obtained through spin coating; the latter were less hydrophilic. However, their water affinity increased with thickness, suggesting that increasing the amount of protein per unit of volume results in more functional groups available to interact with water. Regarding the cast films, the addition of CNC above 10% turned the nanocomposites’ surfaces more hydrophobic. This effect has been widely observed upon addition of highly crystalline nanocellulose crystals, which are known to impart a more hydrophobic character to the films [39], as their interaction with proteins through hydrogen bonding leads to a decrease in the concentration of hydrophilic groups exposed in the film surface [21]. This has also been ascribed to the lotus effect due to the presence of dispersed fillers within the produced films. A similar trend was observed in wheat gluten films [19], reporting that the contact angle increased from 45° to 62° and 75° with 5% and 10% CNC addition, respectively. Atef et al. (2014) also reported higher contact angles (60.8° and 65.6° for 5% and 10% of CNC, respectively) were also reported in agar films [40], while for amaranth protein isolate films made by casting with maize starch nanocrystals, a contact angle increase from 31.4° ± 9.7° to 81.1° ± 8.9° was obtained [21].

Table 3 also compiles the moisture content and solubility of the API films. The moisture content of the films was highly dependent on the processing method. While casting led to water contents of between ~15 and ~21%, spin coating gave rise to much drier materials. In fact, some of the ultrathin API films contained very little moisture left, which also seemed to influence the water solubility of the films. Spin-coated API films had similar values irrespective of the amount of protein, rotational speed, and number of layers, except in the case of the more heterogeneous 15%API film with 100 layers, which showed increased water solubility. In the case of films with 5% of protein, the water solubility ranged from ~2.2 to ~4.3%, which were significantly lower than those reported previously for films with 5% of sunflower protein prepared by casting [41], again highlighting the strong influence of the processing method on the final properties of the materials. In fact, when comparing the water solubility of API films obtained in this work by both methodologies (casting vs. spin coating), a dramatic decrease in solubility was observed for films with the same protein content but obtained through spin coating.

Regarding the effect of CNC incorporation in the cast API films, a decrease in both moisture content and water solubility was observed for the various films developed. The decrease in moisture content could be explained by the stronger H-bond network between API-CNC and CNC-CNC which can reduce the sites available for water sorption. The reported moisture content values of soy protein films with 0–20% starch nanocrystals were between 27 and 33% [20] and for amaranth films with 0–12% maize starch nanocrystals were around 20% [21] and, thus, greater than the ones measured with the materials developed in this work.

Being the an important property of films for food packaging applications, a significant decrease, thus, a broadening of the potential range of application of the materials was observed for the nanocomposite API/CNC films. The decrease in water solubility could be explained by interactions between the amino groups in API with the carbonyl groups of the glucose in CNCs [21].

### 3.3. Mechanical Properties of API Films

The mechanical properties of API and API/CNC films were tested at room temperature, and the results are shown in Table 4. The spin-coated films could not be accurately characterized using the available mechanical testing equipment as, moreover, most of them partially broke when trying to cut the probes and, thus, only the results from the cast films are included in this section.

As observed from Table 4, the neat API film exhibited poor mechanical properties having very low Young modulus and tensile strength values. Similar values have been reported for other proteins like soy protein [42], gelatin [43], and whey protein [35], among others. Both the Young modulus (E-Modulus) and tensile strength of the films significantly increased upon CNC addition. At the same time, elongation at the break was substantially reduced, thus confirming the reinforcing effect of the cellulose nanocrystals, which together with the optical properties suggest good dispersion of the fillers (see also cross-sectional images in Appendix A). This reinforcing effect and interactions between cellulose nanocrystals and proteins had been previously reported by other authors in amaranth [21] and soy-based films [44]. Moreover, as also previously reported [45], the reinforcing effect was concentration-dependent. Other factors, such as the nanocrystals source and affinity with the matrix used, have been seen to strongly affect mechanical properties [21]. From the results obtained and comparing the mechanical properties with those from other biopolymers like thermoplastic starch [46] or synthetic polymers like PET [47], the nanocomposite API films were rather plasticized, displaying significantly lower elastic modulus and greater elongation than the previously mentioned films.

## 4. Conclusions

In this work, two different strategies were compared to improve the properties of amaranth protein isolate (API)-based films; on the one hand, spin-coated films were obtained and a Taguchi experimental design was used to evaluate the influence of protein concentration, spinning rate, and the number of layers on the properties of the films and, on the other hand, bionanocomposites were obtained incorporating different cellulose nanocrystals (CNCs) through a casting methodology. Interestingly, the processing method had a strong influence on some of the properties of the films, like water content, water solubility, and hydrophobicity of the materials. In general, much thinner materials were obtained using spin coating, which could also affect final properties. Good transparency values were obtained for all the materials, although these were also affected by the processing method. In general, spin-coated films displayed reduced solubility and greater hydrophobicity than cast films. In the cast films, an improvement in moisture sensitivity was seen upon CNC addition and the nanocomposites also had better mechanical properties in terms of Young modulus and tensile strength. Therefore, both strategies have been demonstrated to be useful in the improvement of API-based films with potential applications in the food packaging field.

## Figures and Tables

**Figure 1 nanomaterials-10-02564-f001:**
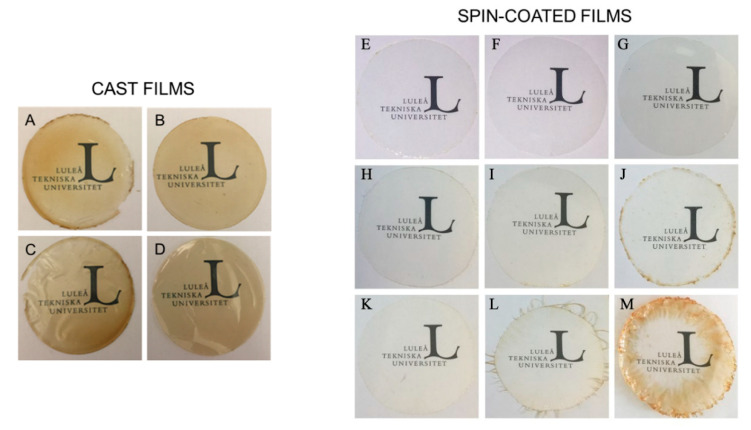
Visual appearance of cast vs. spin-coated amaranth-based films. Cast films: (**A**) API, (**B**) API + CNC 5%, (**C**) API + CNC 10%, (**D**) API + CNC 20% films. Spin-coated films obtained from solutions containing 5% (**E**–**G**), 10% (**H**–**J**) or 15% API (**K**–**M**) and 25 (**E**,**H**,**K**), 50 (**H**–**J**) or 100 layers (**K**–**M**).

**Figure 2 nanomaterials-10-02564-f002:**
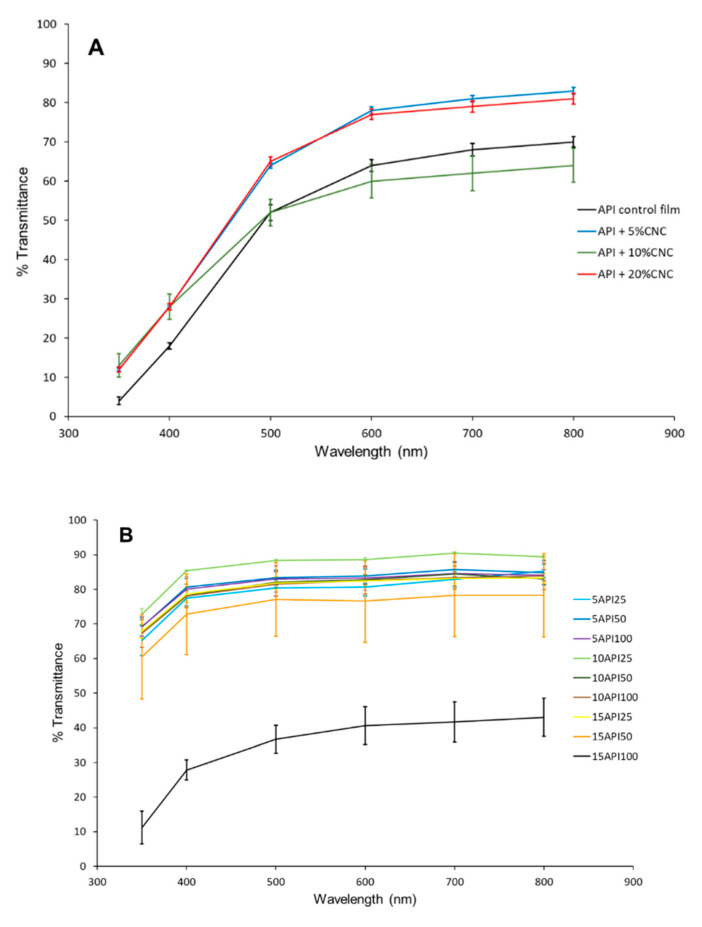
UV-Visible light transmission of the (**A**) cast films and (**B**) spin-coated films.

**Table 1 nanomaterials-10-02564-t001:** Control factors and their levels used for the L_9_ experimental design.

Factor Symbol	Control Factors	Units	Level 1	Level 2	Level 3
A	Protein concentration	wt%	5	10	15
B	Rotational speed	rpm	1000	2500	4000
C	Number of layers		25	50	100

**Table 2 nanomaterials-10-02564-t002:** Orthogonal L_9_ array using the Taguchi method.

Sample Name	Protein Content (%)	Spinning Rate (rpm)	Number of Layers
5API25	5	1000	25
5API50	5	2500	50
5API100	5	4000	100
10API50	10	1000	50
10API100	10	2500	100
10API25	10	4000	25
15API100	15		100
15API25	15		25
15API50	15		50

**Table 3 nanomaterials-10-02564-t003:** Thickness values, transparency, contact angle, moisture content, and water solubility of API films.

Materials	Thickness (µm)	Transparency	Water Contact Angle (°)	Moisture Content (%)	Water Solubility (%)
API cast film	94.1 ± 5.5 ^a^	2.1 ± 0.3 ^b,c^	35.1 ± 1.5 ^c,d^	20.8 ± 0.2 ^a^	42.6 ± 1.9 ^a^
API + 5%CNC	76.3 ± 1.2 ^b^	1.4 ± 0.4 ^c^	31.2 ± 0.8 ^d^	16.9 ± 0.3 ^b^	21.6 ± 0.6 ^c^
API + 10%CNC	80.7 ± 1.6 ^b^	2.7 ± 0.3 ^b^	55.7 ± 2.5 ^b^	14.9 ± 0.1 ^c^	31.4 ± 2.4 ^b^
API + 20%CNC 5API25	85.1 ± 1.2 ^a,b^ 13.2 ± 2.3 ^c,d^	1.3 ± 0.2 ^c^ 7.1 ± 0.8 ^a^	69.9 ± 1.7 ^a^ 56.4 ± 1.8 ^b^	16.1 ± 0.7 ^b,c^ 0.2 ± 0.02 ^h^	37.8 ± 2.7 ^a,b^ 3.4 ± 0.5 ^f^
5API50	7.7 ± 2.2 ^c^	9.9 ± 0.8 ^a^	52.1 ± 1.8 ^b^	1.6 ± 0.35 ^f^	2.2 ± 0.1 ^g^
5API100	9.5 ± 2.4 ^c^	8.4 ± 0.9 ^a^	44.8 ± 2.3 ^c^	0.2 ± 0.03 ^h^	4.3 ± 0.5 ^e^
10API25	18.0 ± 2.9 ^d^	2.9 ± 0.6 ^b^	30.4 ± 2.7 ^d^	0.9 ± 0.06 ^g^	2.7 ± 0.5 ^g^
10API50	29.2 ± 4.1 ^e^	2.8 ± 0.4 ^b^	42.1 ± 1.5 ^c^	0.1 ± 0.06 ^h^	4.0 ± 0.6 ^e,f^
10API100	28.5 ± 2.3 ^d^	2.9 ± 0.7 ^b^	46.8 ± 1.2 ^c^	1.8 ± 0.35 ^e,f^	1.4 ± 0.1 ^h^
15API25	29.3 ± 3.1 ^c^	2.9 ± 0.9 ^b^	35.5 ± 2.6 ^c,d^	2.4 ± 0.71 ^e^	3.3 ± 0.4 ^f,g^
15API50	42.2 ± 5.5 ^f^	2.7 ± 1.5 ^b^	26.9 ± 2.4 ^e^	0.8 ± 0.06 ^g^	4.3 ± 0.2 ^e^
15API100	547.4 ± 21.0 ^g^	0.7 ± 0.5 ^c^	28.9 ± 1.2 ^e^	6.4 ± 0.71 ^d^	16.8 ± 1.7 ^d^

Each value represents mean ± standard deviation (*n* = 5). ^a–h^ different superscripts within the same column indicate significant differences among samples under the Tukey test (*p* < 0.05).

**Table 4 nanomaterials-10-02564-t004:** Mechanical properties of cast API film and its nanocomposites with CNCs.

Samples	E-Modulus (MPa)	Tensile Strength (MPa)	Elongation at Break (%)
API cast film	2.8 ± 0.5 ^a^	0.5 ± 0.1 ^a^	45.7 ± 1.5 ^a^
API + 5%CNC	19.1 ± 2.8 ^b^	2.8 ± 0.5 ^b^	26.8 ± 1.3 ^b^
API + 10%CNC	55.1 ± 1.8 ^c^	4.1 ± 0.6 ^c^	22.9 ± 0.8 ^c^
API + 20%CNC	79.9 ± 6.0 ^d^	5.3 ± 0.1 ^d^	13.9 ± 0.7 ^d^

Each value represents mean ± standard deviation (*n* = 5). ^a–d^ different superscripts within the same column indicate significant differences among samples under the Tukey test (*p* < 0.05).

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
