# Peer review of "Strategies to Improve the Properties of Amaranth Protein Isolate-Based Thin Films for Food Packaging Applications: Nano-Layering through Spin-Coating and Incorporation of Cellulose Nanocrystals"

_nanomaterials, 2020, doi:10.3390/nano10122564_

Round 1

Reviewer 1 Report

Although I am not an expert in the field of biopolymers, I support my positive assessment in my previous review with a favorable assessment of the first version of manuscript. Of course, a revised version of manuscript - taking into account the insightful comments of other reviewers - is much better, as it explains many of the technological details of the methods used, and explains a number of doubts. In particular, I consider it valuable to determine the front lines of both strategies used to produce the layers of nano-biocomposites: Both lead to a reduction in the hydrophilic character of the films.

I believe that, after taking into account valuable comments/comments from reviewers, this manuscript deserves to be published in Nanomaterials MPDI. I think that any progress in improving the packaging technology of foods using biopolymers is worth publishing, as it will allow environmentally harmful PET packaging in the future.

Author Response

Thanks for the positive comments about our reviewed version

Reviewer 2 Report

The resubmitted paper reports on two different strategies for the development of amaranth protein isolate based films, being focused on a very interesting subject. The paper can be accepted in the current version. It is suggested to add as last keyword the potential application, i.e. ‘food packaging’.

Author Response

Ok, we will include food packaging as last keyword in the manuscript. Thanks for the comments.

Reviewer 3 Report

The publication entitled "Strategies to Improve the Properties of Amaranth Protein Isolate-Based Thin Filmsfor Food Packaging Applications: Nano-layering through Spin-Coating and Incorporation of Cellulose Nanocrystals" which I am reviewing for the second time has one single common denominator with nanomaterials in the form of CNCs additive. It would be reasonable to check whether this component is still nano in the finished material. As I wrote before, SEM imaging would be enough. There are also no measurements allowing to determine the chemical properties of the material. (I wrote about it in my earlier review and the authors did not take it into account.) The article is not badly written from a scientific point of view, but the question arises whether publishing it in this journal is justified in terms of the subject/topic. This is the biggest problem, but I believe it is up to the editor to judge it. These facts do not disqualify work as far as publications is concerned but in this case, the final decision in my opinion should be made by the editor.

Author Response

See responses attached

This manuscript is a resubmission of an earlier submission. The following is a list of the peer review reports and author responses from that submission.

Round 1

Reviewer 1 Report

In my opinion in publication entitled “Strategies to Improve the Properties of Amaranth Protein Isolate-Based Thin Filmsfor Food Packaging Applications: Nano-layering through Spin-Coating and Incorporation of Cellulose Nanocrystals” nano aspect are insufficiently developed. In addition, items of novelty are missing. The observation that spin coater creates thin film that are different then  classic ones poured to mold is not new. The author states that thickness is one of difference, although it was possible to try to obtain materials of similar thickness which would make it easier to compare other important parameters.  The authors writes that they took pictures using an electron microscope but does not show results. Such characterization should be performed and it is critical to the assessment of this material nano scale morphologies . The chemical characteristics of the material is missing, it is not known if the method of production affects the chemical composition and distribution of the component of the mixture. The topic itself is very interesting and important but the work requires a lot of refinement. Additional measurements must be made, this work cannot be published at this stage.

Author Response

See reply in the document attached

Reviewer 2 Report

General comments

The submitted manuscript proposes two different strategies for the development of amaranth protein isolate based films: the first one was based on the production of ultrathin films by spin coating nanolayering ,and the second one on the addition of CNCs to API films by means of casting process.

The topic is interesting and worthy of investigations, but major revisions have to be applied before considering this paper for possible publication in Nanomaterials.

It is not clear why they did not prepare the CNCs loaded samples by spin coating. The Authors have to justify this point in the Introduction or Materials and Methods section.

The originality of the present work has to be evidenced and the results have to be better discussed and compared with the literature.

Moreover, a revision of English grammar and language is strongly recommended.

More details and specific suggestions are reported below point by point.

 Keywords

The chosen keywords (i.e.  Amaranth protein isolate; cellulose nanocrystals; spin coating; casting; thin films) do not completely cover the review content. Further ones have to be added, such as those regarding specific properties and the potential application.

  1. Introduction
  • The incipit from “The use of biopolymers, such as polysaccharides, lipids or proteins, has generated growing interest in recent years for their potential application as edible films and coatings for use in food packaging and to counteract the problem of pollution due to the use of plastic packaging [1]” has to be supported with more recent references, including “Eco-sustainable systems based on poly(lactic acid), diatomite and coffee grounds extract for food packaging, Intern J Biolog Macromolecules 112, June 2018: 567-575”.
  • The following statements “Various types of proteins such as wheat gluten, soy, gelatin, corn zein, peanut, sun flower, milk and whey proteins have been evaluated as potential film forming agents [2-3]. Although the developed materials have good barrier properties to oxygen, flavors and lipids, they show high water vapor permeability and poor mechanical properties [3]” have to be corroborated with more recent suitable references, including “Biodegradable zein film composites reinforced with chitosan nanoparticles and cinnamon essential oil: physical, mechanical, structural and antimicrobial attributes, Colloids and Surfaces B: Biointerfaces 177(2019): 25-32.” .
  • The following sentence “Natural polysaccharides, such as nano sized cellulose, chitin and starch, can be employed as nanoreinforcing elements because of their high mechanical properties” needs appropriate references, including “Effect of silver nanoparticles and cellulose nanocrystals on electrospun poly(lactic) acid mats: morphology, thermal properties and mechanical behaviour, Carbohydrate Polymers 103 (2014): 22– 31”.
  • It is not clear why they did not prepare the CNCs loaded samples by spin coating. The Authors have to justify this point in the Introduction or Materials and Methods section.
  • The Authors should better evidence the originality of their work and the added value to the scientific knowledge about the considered topic.
  • It is strongly suggested to add a brief list of the used characterisations at the end of the Introduction section.

  1. Materials and Methods

- The Authors have to justify whythey decided to prepare the CNCs loaded samples by solvent casting and not by spin coating.

2.1. Materials

- Even if reported elsewhere, the protocol followed for the preparation of the amaranth protein isolate has to be briefly described.

- More details about all the used reagents have to be added, particularly for the commercial amaranth protein concentrate and the used CNCs.

2.2. Preparation of the API solutions

- Were the solutions prepared in pure acid formic or in an acid formic solution? It has to be specified, indicating the formic acid concentration.

2.3. Nano-layering through spin coating process

- The set up coating time for the spin coating process has to be specified. Moreover the chosen number of layers has to be justified.

2.4. Casting

- Please, specify that the used procedure is ‘solvent casting’.

  • The diameter of the polystyrene Petri dishes has to be reported.

2.5. Characterization

- The UV/Vis Spectrometer  resolution has to be reported.

- A reference for the measurement of film opacity has to be added.

- A reference for the determination of the Moisture content  has to be added.

- For the mechanical tests, the followed standard has to be specified.

  1. Results and Discussion

3.1. Optical properties

As a general consideration, the Authors cannot compare the spin coated samples with the cast ones loaded with CNCs. They can only compare the spin coated samples and cast samples obtained starting from the same concentration of API solution (5%).

3.1 Synthesis and Impacts of Hydrothermal Annealing on Crystallite Size

- The following consideration “The exceptions were the spin-coated films obtained from the 15% protein solution with 50 and 100 layers and, especially, the latest one which was obtained at low spinning rate, and showed a rather heterogeneous appearance and varying thickness, as less material was deposited in the center of the film due to the centrifugal force during the film forming process” has to be supported with suitable references.

- Moreover the reported solvent-cast and spin coated samples are not so comparable, since they were produced starting from CNCs suspensions in the first case and CNCs free solutions in the second case.

- The Authors stated that “greater light transmission values were observed for the spin-coated films in the whole spectra. It is interesting to note the ability of cast films to block UV light, which could be useful for certain applications, like the packaging of light-sensitive products”. It is obvious since the cast films contained CNCs, whereas the spin coated ones did not contain fillers.

- Similarly, the following comparison “Interestingly, when comparing the films prepared by casting and spin-coating having the same protein concentration (5%), the latter ones displayed greater transparency values, indicating that this processing technology may lead to more transparent films” has no sense, since the starting solutions/suspensions were not the same.

3.2. Moisture sensitivity of the API films

- The increment of the contact angle with the CNCs amount has to be also ascribed to the ‘loto effect’ due to the presence of dispersed fillers within the produced films.

- The following consideration “In fact, when comparing the water solubility of API films obtained in this work by both methodologies (casting vs. spin coating), a dramatic decrease in solubility was observed for films with the same protein content but obtained through spin coating” has to be justified and supported with proper references.

- The following conclusion “The decrease in water solubility could be explained by interactions between the amino groups in API with the carbonyl groups of the glucose in CNCs.” has to be corroborated with appropriate references.

Author Response

See reply in attached document

Reviewer 3 Report

The manuscript describes two strategies (spin coating & casting) leading to improve the properties of Amaranth based films with the potential use for food packaging applications. The main conclusion is that both strategies result in homogeneous optically transparent films with improved mechanical properties and hydrophobicity when compared with the neat Amaranth films. The methodology is described clearly with a lot of details. I find the manuscript worth publication in Nanomaterials. Remarks I find the optical properties less important than the others (mechanical, hydrophobicity, solubility etc,). Therefore, it would be valuable to compare such properties with the same properties of the polymeric materials (unfortunately) commonly used in packaging.

Author Response

See reply in attached document

Round 2

Reviewer 1 Report

The authors do not address the problematic issues in a way that would satisfy me. The quality of work has only improved only slightly (SEM photos are the added value). It is not enough to be published. The spin coater has been known in pure science for over 20 years it is hard to talk about novelty here. If the authors showed how to use this method to obtain very large sheets of material that could be realistically used in industry, it would be a novelty. Then we would have, for example, a description of the construction of a new apparatus or something like that. We don't have anything like that in this case. The author does not want to perform the chemical characteristics of the material, writing that it is in an earlier work. Again where the news. Pouring and heating the material may cause it to oxidize for synthesis with spin coater it won't happen. The starting material is colloidal spin coater centrifugal force can lead to uneven distribution of colloidal particles. The author does not want to address these topics. Nowadays, a work to be published must contain a strong element of novelty or a lot of results/use many different measurement methods, and it is not here.

Author Response

We think that the reviewer has not understand the novelty of the presented work. Again, although spin coating has been known for a very long time, it has been mainly used in electronic applications and only scarcely used for processing biological materials like proteins. We completely agree with the reviewer in that we are not presenting any new advance regarding the technique, but we would like to highlight that it was not the aim of this work to present the technique for industrial application. This is an exploratory work which aims to understand how different film-forming techniques (spin coating vs. casting) affect the properties of the protein-based materials.

Regarding the chemical characteristics of the materials, as previously stated, there is no point in characterizing them, as both solutions were processed exactly the same and the only difference was that one of the solutions was left for solvent evaporation (casting), while the other was spin coated.

Reviewer 2 Report

General comments

The authors did not follow all the referee’s suggestions. Thus, some minor revisions have to be applied.

 Keywords

The chosen keywords (i.e.  Film processing; edible protein; packaging; cellulose nanocrystals; thin films) are not proper. The previous ones (i.e. Amaranth protein isolate; cellulose nanocrystals; spin coating; casting; thin films) sounded better, even if it was suggested to add further ones, such as those regarding specific properties and the potential application. Moreover, a logical order (i.e. material, processing, characterisation, properties, applications) has to be applied.

2.1. Materials

- The Authors did not add more details about all the used reagents, particularly for the commercial amaranth protein concentrate and the used CNCs, as requested in the previous review.

Author Response

Regarding the comment about the keywords, we have modified them as requested following the order suggested and we have included an extra one about characterization. All the changes are in red font to facilitate revision.

Regarding the materials, more details about the amaranth protein isolate used and the CNCs have been provided as requested